# A Genetically-Informed Network Model of Myelodysplastic Syndrome: From Splicing Aberrations to Therapeutic Vulnerabilities

**DOI:** 10.3390/genes16080928

**Published:** 2025-08-01

**Authors:** Sanghyeon Yu, Junghyun Kim, Man S. Kim

**Affiliations:** 1Translational-Transdisciplinary Research Center, Clinical Research Institute, Kyung Hee University Hospital at Gangdong, College of Medicine, Kyung Hee University, Seoul 05278, Republic of Korea; sanghyeon99@khu.ac.kr; 2Department of Biomedical Science and Technology, Graduate School, Kyung Hee University, Seoul 02453, Republic of Korea; 3Division of Tourism & Wellness, Hankuk University of Foreign Studies, Yongin-si 17035, Republic of Korea; jh.kim@hufs.ac.kr

**Keywords:** myelodysplastic syndrome, stem cell, splicing mutations, epitranscriptomics, precision medicine

## Abstract

**Background/Objectives**: Myelodysplastic syndrome (MDS) is a heterogeneous clonal hematopoietic disorder characterized by ineffective hematopoiesis and leukemic transformation risk. Current therapies show limited efficacy, with ~50% of patients failing hypomethylating agents. This review aims to synthesize recent discoveries through an integrated network model and examine translation into precision therapeutic approaches. **Methods**: We reviewed breakthrough discoveries from the past three years, analyzing single-cell multi-omics technologies, epitranscriptomics, stem cell architecture analysis, and precision medicine approaches. We examined cell-type-specific splicing aberrations, distinct stem cell architectures, epitranscriptomic modifications, and microenvironmental alterations in MDS pathogenesis. **Results**: Four interconnected mechanisms drive MDS: genetic alterations (splicing factor mutations), aberrant stem cell architecture (CMP-pattern vs. GMP-pattern), epitranscriptomic dysregulation involving pseudouridine-modified tRNA-derived fragments, and microenvironmental changes. Splicing aberrations show cell-type specificity, with SF3B1 mutations preferentially affecting erythroid lineages. Stem cell architectures predict therapeutic responses, with CMP-pattern MDS achieving superior venetoclax response rates (>70%) versus GMP-pattern MDS (<30%). Epitranscriptomic alterations provide independent prognostic information, while microenvironmental changes mediate treatment resistance. **Conclusions**: These advances represent a paradigm shift toward personalized MDS medicine, moving from single-biomarker to comprehensive molecular profiling guiding multi-target strategies. While challenges remain in standardizing molecular profiling and developing clinical decision algorithms, this systems-level understanding provides a foundation for precision oncology implementation and overcoming current therapeutic limitations.

## 1. Introduction

Myelodysplastic syndrome (MDS) encompasses a diverse spectrum of clonal hematopoietic stem cell (HSC) malignancies that primarily affect elderly individuals and is characterized by the paradox of increased bone marrow cellularity alongside peripheral blood cytopenia, morphologic dysplasia, and a substantial risk of leukemic transformation [1,2]. This fundamental contradiction, abundant yet ineffective hematopoiesis, has long epitomized the biological complexity of MDS and underscored the challenges in developing effective therapeutic interventions.

The clinical heterogeneity of MDS has historically posed significant obstacles to accurate diagnosis, meaningful risk stratification, and rational therapeutic selection. Current treatment approaches have largely relied on hypomethylating agents (HMAs) as the therapeutic backbone [3,4], yet approximately 50% of patients fail to achieve any meaningful responses, and those who initially respond eventually develop resistance mechanisms that lead to treatment failure [5,6]. This therapeutic plateau highlights the urgent need for novel strategies based on deeper mechanistic understanding.

The past three years have witnessed an unprecedented convergence of technological innovations and biological insights that have fundamentally reshaped our understanding of MDS pathogenesis. Single-cell multi-omics technologies have revealed that the consequences of common genetic alterations are far more cell-type-specific and context-dependent than previously appreciated [7,8]. Simultaneously, the identification of distinct (HSC) architectures has provided a biological framework for understanding why patients with seemingly similar disease characteristics can demonstrate substantially different treatment responses [9].

Most significantly, emerging evidence has suggested that these various molecular mechanisms do not operate in isolation but form an interconnected network where splicing aberrations influence stem cell differentiation programs and epitranscriptomic modifications, shape therapeutic vulnerabilities and microenvironmental alterations, and drive resistance patterns [10]. This system-level understanding is beginning to enable truly precise therapeutic approaches, moving beyond single-biomarker-guided decisions toward comprehensive molecular profiling to guide multitarget intervention strategies.

This review synthesizes recent discoveries in MDS through an integrated network model, departing from previous analyses that treated molecular pathways in isolation. We propose that four key mechanisms—genetic alterations (notably splicing factor mutations), aberrant stem cell architecture, epitranscriptomic dysregulation, and microenvironmental changes—form an interconnected network that collectively drives MDS pathogenesis (Figure 1A) [7,9,10,11]. This systems-level understanding, built on breakthroughs from the past three years, provides the rationale for transitioning from single-biomarker therapies to multitarget strategies guided by comprehensive molecular profiling [12,13]. Critically, we also outline a framework for overcoming clinical implementation challenges, aiming to translate these complex insights into actionable treatment algorithms.

Central to this integrated view is how distinct stem cell architectures define the heterogeneity of MDS and predict therapeutic responses (Figure 1B). We provide a unifying framework based on two contrasting architectural patterns: a common myeloid progenitor (CMP) pattern and a granulocyte–monocyte progenitor (GMP) pattern [9]. The CMP pattern is characterized by an expansion of CMPs and is frequently associated with TP53 mutations, whereas the GMP pattern shows expanded GMPs and is linked to mutations such as RUNX1 and DNMT3A [14,15,16]. Understanding these architectural differences is not merely descriptive; it is crucial for predicting clinical outcomes and guiding modern therapeutic strategies, including venetoclax-based therapies [12,13].

## 2. Interconnected Molecular Mechanisms in MDS Pathogenesis

### 2.1. Single-Cell Multi-Omics Reveals the Cellular Context of Splicing Aberrations

The development of Genotyping of Transcriptomes with Splicing (GoT-Splice) technology has fundamentally transformed our understanding of how splicing factor mutations contribute to MDS pathogenesis by revealing their cell-type specificity [7]. Rather than causing uniform splicing defects across all hematopoietic lineages, these mutations demonstrate remarkable selectivity for their cellular targets and functional consequences.

Previous studies employing GoT-Splice in patients with SF3B1-mutated MDS have revealed that mutant cells demonstrate preferential enrichment within the megakaryocytic–erythroid lineage, with significant expansion, specifically within erythroid progenitor populations [8,17]. This selective lineage bias has provided mechanistic insight into the characteristic clinical presentation of SF3B1-mutated MDS, which typically manifests as macrocytic anemia with ring sideroblasts [18]. Critically, these investigations have uncovered distinct patterns of cryptic 3′ splice site usage that vary not only between the different progenitor populations but also across different stages of cellular maturation within the same lineage [19].

Long-read single-cell transcriptomics has revealed that SF3B1-mediated splicing aberrations intensify progressively as cells advance through erythroid differentiation, directly correlating with the accumulation of dysfunctional mitochondria and the formation of pathognomonic ring sideroblasts [20]. This stage-specific intensification suggests that the cellular consequences of splicing mutations are amplified by the increasing metabolic demands of terminal differentiation, creating a positive feedback loop between splicing defects and cellular dysfunction.

Recent studies have extended beyond SF3B1 in order to examine U2AF1, SRSF2, and ZRSR2 mutations, revealing that each splicing factor has unique cellular preferences and distinct downstream consequences [21,22,23,24]. SRSF2 mutations have been associated with specific mitochondrial dysfunction and altered mitophagy pathways [22], whereas U2AF1 mutations demonstrate distinct patterns of alternative 3′ splice site selection that affect different sets of target genes [23]. These findings suggest that the specific splicing factors involved may influence not only the clinical phenotype but also the underlying therapeutic approach.

### 2.2. Stem Cell Architecture as a Unifying Framework for Disease Heterogeneity

The identification of two distinct HSC architectures—CMP-pattern and GMP-pattern MDS—represents one of the most significant conceptual advances in MDS research [9]. These architectures provide a biological framework for understanding the heterogeneity of disease presentation, progression patterns, and treatment responses, which has long challenged clinicians and researchers.

CMP-pattern MDS is characterized by an increased frequency of common myeloid progenitors within the hematopoietic hierarchy and is associated with higher frequencies of long-term HSCs and multipotent progenitors [9]. The molecular underpinnings of this architecture involve the preferential activation of specific transcriptional programs that maintain stemness while promoting myeloid commitment. In contrast, GMP-pattern MDS demonstrates increased granulocytic–monocytic progenitor frequencies and significant expansion of lymphoid-primed multipotent progenitors, reflecting a fundamentally different differentiation trajectory [9].

The genetic landscape underlying these distinct architectures provides crucial insights into disease biology. TP53 mutations are significantly enriched in CMP-pattern MDS [14,25], whereas RUNX1, DNMT3A, BCOR, and STAG2 mutations are more prevalent in GMP-pattern MDS [15,16]. These genetic associations are not merely correlative; they appear to drive distinct cellular architectures through differential effects on stem cell self-renewal, differentiation capacity, and survival pathway activation.

Although specific splicing mutations exhibit a clear lineage bias, the mechanistic link explaining how these cell-type-specific effects directly contribute to the formation of distinct CMP- or GMP-patterned stem cell architectures remains poorly understood and represents a critical area for future investigation.

### 2.3. Epitranscriptomic Regulation: A Novel Layer of Control

The discovery of epitranscriptomic mechanisms in MDS pathogenesis has revealed an entirely new layer of post-transcriptional regulation that operates independently of genetic and epigenetic alterations but potentially intersects [10]. Pseudouridine-modified transfer RNA-derived fragments (tRFs), particularly mini tRFs containing a 5′ terminal oligoguanine (mTOG) motif, have emerged as critical regulators of protein synthesis in hematopoietic stem and progenitor cells [10,26].

Past mechanistic studies have demonstrated that pseudouridylation of mTOG by the synthase PUS7 enables selective binding to polyadenylate-binding protein cytoplasmic 1 (PABPC1), resulting in the destabilization of the translation–initiation complex eIF4F [10,27]. This interaction specifically represses translation of transcripts containing pyrimidine-enriched sequences at the 5′ untranslated region, including 5′ terminal oligopyrimidine tracts that encode the essential protein machinery components frequently altered in cancer [27].

The depletion of mTOG in MDS leads to aberrant upregulation of protein synthesis, particularly affecting transcripts encoding ribosomal proteins and translation initiation factors [10]. This dysregulation appears to be most pronounced in high-risk MDS and is correlated with impaired hematopoietic differentiation and increased transformation risk. Importantly, low mTOG levels serve as powerful prognostic indicators that appear to operate independently of genetic factors, suggesting that epitranscriptomic alterations may provide complementary information to existing biomarkers [10].

Although low mTOG levels clearly serve as powerful prognostic indicators that appear to operate independently of genetic factors, a deeper “integrated understanding” will require investigation of whether these epitranscriptomic alterations are driven by specific genetic mutations (e.g., SRSF2 and TET2) or are associated with distinct stem cell architectures, such as the CMP or GMP pattern.

### 2.4. Microenvironmental Alterations: The Supporting Cast Becomes Central

Recent investigations revealed that MDS involves complex alterations in the bone marrow microenvironment that extend far beyond malignant clones [11,28]. These microenvironmental changes appear to respond to and actively promote disease progression, creating a supportive niche for malignant stem cells while simultaneously impairing normal hematopoiesis.

TET2 mutations are among the most frequent genetic alterations in MDS, demonstrating the complex interplay between malignant cells and their microenvironments [29,30,31]. TET2 mutations affect not only malignant myeloid cells but also natural killer (NK) cells, resulting in compromised antileukemic immune surveillance [29]. Patients with MDS harboring TET2 mutations exhibit phenotypic and functional defects in circulating NK cells, including increased DNA methylation, reduced expression of killer immunoglobulin-like receptors, and decreased cytotoxic protein expression [29].

The bone marrow stromal microenvironment in MDS demonstrates the significant alterations in specific cell populations, cytokine networks, and extracellular matrix composition [11,28]. These changes create a protective niche for malignant stem cells through multiple mechanisms, including enhanced survival signaling, metabolic support, and physical protection from therapeutic agents. Recent studies have identified specific stromal cell populations that actively promote MDS cell survival through direct cell–cell contact and paracrine signaling involving growth factors such as IL-3, GM-CSF, and SCF [11,32].

### 2.5. Mechanistic Interconnections: A Systems-Level Perspective

The molecular mechanisms described above do not operate independently but form a complex, interconnected network where perturbations in one pathway cascade through multiple others. Understanding these interconnections is crucial for developing effective therapeutic strategies and predicting treatment responses.

Splicing factor mutations demonstrate context-dependent effects that vary significantly based on the cellular context [7]. SF3B1-mutated cells demonstrate preferential enrichment within the megakaryocytic–erythroid lineage, with significant expansion specifically within erythroid progenitor populations [8,17]. Long-read single-cell transcriptomics has revealed that SF3B1-mediated splicing aberrations intensify progressively as cells advance through erythroid differentiation, directly correlating with the accumulation of dysfunctional mitochondria and the formation of pathognomonic ring sideroblasts [20]. However, the mechanistic link explaining how these cell-type-specific effects directly contribute to the formation of distinct CMP- or GMP-patterned stem cell architectures remains poorly understood and represents a critical area for future investigation.

Epitranscriptomic alterations appear to operate as a regulatory layer that can modulate the cellular consequences of both genetic mutations and stem cell architectural changes [10,27]. Pseudouridine-modified transfer RNA-derived fragments (tRFs), particularly mini tRFs containing a 5′ terminal oligoguanine (mTOG) motif, have emerged as critical regulators of protein synthesis in hematopoietic stem and progenitor cells [10]. Low mTOG levels not only serve as independent prognostic indicators but may also influence how cells respond to therapeutic interventions [10]. Although low mTOG levels clearly serve as powerful prognostic indicators that appear to operate independently of genetic factors, a deeper “integrated understanding” will require investigation of whether these epitranscriptomic alterations are driven by specific genetic mutations (e.g., SRSF2 and TET2) or are associated with distinct stem cell architectures, such as the CMP or GMP pattern [10].

Microenvironmental alterations both respond to and actively promote the other molecular changes [11,29]. TET2 mutations are among the most frequent genetic alterations in MDS, demonstrating the complex interplay between malignant cells and their microenvironments [29]. TET2 mutations affect not only malignant myeloid cells but also natural killer (NK) cells, resulting in compromised antileukemic immune surveillance [29]. Patients with MDS harboring TET2 mutations exhibit phenotypic and functional defects in circulating NK cells, including increased DNA methylation, reduced expression of killer immunoglobulin-like receptors, and decreased cytotoxic protein expression [29].

These interconnections have profound therapeutic implications. Single-target approaches often fail because they address only one component of a multi-layered pathogenic network, as evidenced by the fact that approximately 50% of patients fail to achieve meaningful responses to hypomethylating agents [5,6]. Successful therapeutic strategies must account for these interconnections, either by simultaneously targeting multiple pathways or by identifying key nodal points where the intervention can disrupt multiple pathogenic mechanisms.

Table 1 summarizes how the four key mechanisms discussed—splicing factor mutations, stem cell architecture, epitranscriptomic regulation, and microenvironmental alterations—function as an integrated network. For instance, the table clarifies that SF3B1 mutations specifically impact erythroid progenitors, leading to anemia with ring sideroblasts, whereas TP53 mutations are associated with a CMP-pattern stem cell architecture. Thus, each mechanism is linked to distinct cellular and clinical features that explain disease heterogeneity and form the basis for precision therapeutic strategies.

## 3. Precision Therapeutic Strategies: From Mechanism to Medicine

### 3.1. Biomarker-Guided Treatment Selection: Venetoclax and Stem Cell Architecture

The translation of mechanistic insights into clinical practice has been successfully achieved through the development of stem cell architecture-based treatment selection for venetoclax therapy [12,13]. This represents one of the first examples of precision medicine in MDS, where biological understanding directly informs therapeutic decision-making.

Clinical studies have demonstrated that patients with CMP-pattern MDS achieve markedly superior outcomes with venetoclax combination therapy, with response rates exceeding 70% compared to less than 30% in patients with GMP-pattern MDS (Figure 2A) [12,34]. The biological basis for this differential response lies in the preferential activation of BCL-2-mediated anti-apoptotic pathways in CMP-patterned HSCs during disease progression, creating a targetable vulnerability [33,35]. The therapeutic implications of this classification system have proven to be profound, with clinical validation confirming that patients with CMP-pattern MDS achieve superior responses to venetoclax-based therapy, including a shorter time to complete remission and significantly longer relapse-free survival, compared to patients with GMP-pattern MDS [12,13].

However, the clinical reality is often more complex than single-biomarker-guided decisions. The presence of TP53 mutations appears to override other potentially favorable biomarkers, significantly diminishing treatment efficacy regardless of the stem cell architecture pattern [25,36,37]. Clinical data demonstrate that although venetoclax combinations show promise in certain genetic subsets, the presence of TP53 mutations creates a dominant-negative prognostic factor that limits treatment success. In patients with TP53-mutated MDS and acute myeloid leukemia, the burden of TP53 aberrations (high-risk vs. low-risk conditions based on the allelic state) significantly affects outcomes, with biallelic TP53 alterations associated with a poor prognosis [25].

### 3.2. Venetoclax Combinations and Rational Drug Development

The success of biomarker-guided venetoclax therapy has prompted the investigation of more sophisticated combination strategies that can simultaneously address multiple pathways [12,13,38]. The combination of venetoclax with hypomethylating agents has demonstrated remarkable efficacy in specific patient subsets, leading to investigations across a spectrum of MDS risk categories [39,40,41].

Recent clinical trials have established the safety and efficacy of venetoclax plus azacitidine in patients with higher-risk MDS, particularly in those in whom prior HMA therapy failed [42,43]. However, these benefits are largely restricted to patients with specific biological characteristics, reinforcing the importance of patient selection. Optimal dosing strategies have evolved from extended initial exposure schedules to more refined approaches that minimize myelosuppression while maintaining efficacy [42].

The development of rational combination regimens represents the next frontier of precision combination therapy. These approaches simultaneously target malignant clones, overcome resistance mechanisms, and modify supportive microenvironments. Promising combinations under investigation include venetoclax plus HMAs with FLT3 inhibitors for FLT3-mutated diseases [44] and novel agents targeting microenvironmental support mechanisms [45,46].

### 3.3. Understanding and Overcoming Treatment Resistance

Understanding the mechanisms underlying HMA resistance has become increasingly critical as these agents remain the therapeutic foundation for most patients with MDS [47,48,49]. The molecular basis of resistance involves complex mechanisms including intrinsic cellular properties and adaptive responses that evolve during treatment.

At the stem cell level, HMA therapy fails to eliminate the most primitive disease-initiating cells, which maintain their self-renewal capacity and have the ability reconstitute the malignant clone following treatment [48,49]. Studies using mouse models have demonstrated that while HMA treatment effectively depletes differentiated progenitor cells, primitive HSCs survive treatment and aberrantly reconstitute downstream compartments [48]. Surviving stem cells maintain their characteristic disease phenotypes and clonal burdens, thus providing a reservoir for disease persistence and progression.

The role of the microenvironment in mediating treatment resistance is increasingly being recognized as a critical factor that may require specific therapeutic targeting [11,32]. Resistant stem cells appear to be preferentially located within specific microenvironmental niches that provide protection from therapeutic agents through multiple mechanisms, including enhanced survival signaling, altered drug metabolism, and physical barriers to drug penetration.

Combination strategies designed in order to address microenvironment-mediated resistance include agents that disrupt stem cell–niche interactions, modify stromal cell function, and enhance the immune-mediated elimination of resistant cells. These approaches may be particularly important for achieving deep and durable responses that eliminate disease-initiating cell populations.

Further investigations are needed to determine whether primitive disease-initiating stem cell clones that survive HMA therapy preferentially exhibit a CMP- or GMP-pattern architecture. If such a link is established, it could serve as a valuable biomarker for identifying patients with intrinsic resistance to HMA treatment.

Table 2 provides a comprehensive analysis of the multifaceted nature of this therapeutic resistance. It systematically categorizes how resistance emerges at multiple levels, including the survival of primitive disease-initiating stem cells, the emergence of specific genetic mutations, and the role of the protective microenvironment. Furthermore, by contrasting current standard-of-care treatments with emerging strategies designed to overcome these resistance mechanisms, the table highlights the limitations of single-agent therapies and underscores the necessity for combination treatments that target multiple pathways simultaneously.

### 3.4. Novel Targets and Emerging Therapeutic Approaches

The expanding understanding of MDS biology has revealed numerous novel therapeutic targets that offer potential alternatives to conventional approaches [21,55,56]. Splicing factor mutations, present in up to 60% of patients with MDS, represent particularly attractive targets for therapeutic intervention.

Recent drug development efforts have focused on compounds that modulate the activity of mutant splicing factors and correct their downstream effects [19,21]. These include modulators of the U2 snRNP complex, inhibitors of specific protein–protein interactions within the spliceosome, and compounds that can restore normal splicing patterns in cells harboring splicing factor mutations. Early preclinical studies have suggested that these approaches can induce differentiation and apoptosis in MDS cells while sparing normal hematopoietic cells [19,57].

Protein arginine methyltransferase 5 (PRMT5) has emerged as a promising therapeutic target through the principle of synthetic lethality with certain splicing factor mutations [55]. PRMT5 inhibition demonstrated selective toxicity in cells harboring specific splicing factor mutations, providing a therapeutic window for targeting malignant cells while sparing normal hematopoietic cells. A recent phase 1 clinical trial of the PRMT5 inhibitor JNJ-64619178 in patients with lower-risk MDS demonstrated target engagement and tolerability, although its clinical efficacy was found to be limited [58]. Despite the lack of objective responses in this initial study, the concept of targeting PRMT5 in splicing factor-mutated MDS remains promising and warrants further investigation with optimized dosing strategies and patient selection criteria.

The epitranscriptomic pathway involving PUS7 and mTOG represents a novel therapeutic opportunity to address fundamental disease mechanisms [10,59]. Although strategies targeting this pathway are under development, significant challenges related to delivery efficiency, stability, and potential off-target effects must be addressed before clinical translation.

The expanding understanding of MDS biology has revealed numerous novel therapeutic targets that offer potential alternatives to conventional approaches, as summarized in our integrated therapeutic framework (Figure 2B) [21,55,56].

## 4. Advanced Diagnostics and Clinical Implementation

### 4.1. Comprehensive Molecular Profiling for Precision Diagnosis

The integration of advanced genomic technologies has significantly enhanced the diagnostic accuracy and prognostic precision of MDS, moving beyond conventional cytogenetics toward comprehensive molecular characterization [53,60,61]. Optical genome mapping (OGM) has emerged as a powerful tool for detecting cryptic chromosomal abnormalities missed by standard approaches, including therapeutically relevant fusion genes with non-canonical breakpoints [62].

Recent applications of OGM have identified cryptic rearrangements, such as CBFB::MYH11 fusions with unusual breakpoints that are not detected by standard molecular techniques, highlighting critical diagnostic blind spots in current approaches [62]. These findings underscore the potential of genome-wide technologies to identify actionable alterations that could guide treatment decisions.

Whole-exome sequencing has revealed novel prognostic mutations beyond the traditional driver genes, including mutations in genes that serve as independent risk factors for survival [63,64]. The identification of these unexpected prognostic markers demonstrates the value of comprehensive genomic profiling and suggests that our current understanding of the relevant genetic alterations in MDS remains incomplete.

Single-cell genomics technologies enable the tracking of individual clones over time, the identification of rare-resistant subpopulations, and the characterization of evolutionary pressures that drive disease progression [51,52]. Understanding clonal architecture has important implications for predicting treatment responses and identifying patients at a high risk of rapid transformation [54,65].

### 4.2. Molecular Risk Stratification and Prognostic Integration

The integration of molecular biomarkers into prognostic scoring systems represents a major advancement in regard to MDS risk stratification, with molecular scoring systems demonstrating improved accuracy when compared to purely morphological and cytogenetic approaches [53,61].

TP53 mutation status has emerged as one of the most powerful prognostic indicators, with allelic status providing additional refinement [14,25,66]. Patients with biallelic TP53 alterations demonstrate particularly poor outcomes and limited responses to conventional therapies, making them candidates for investigational approaches or early consideration of allogeneic transplantation [50,67].

GATA2 mutations have been identified as significant prognostic factors, particularly in younger patients with familial predisposition [68,69]. Recognition of GATA2 deficiency syndrome has important implications for family screening and consideration of early interventions. DDX41 mutations represent another increasingly recognized entity associated with improved overall survival compared with other genetic subtypes [34,70].

The integration of multiple molecular markers into comprehensive prognostic models remains challenging because different markers may provide conflicting prognostic information. Developing algorithms that can appropriately weigh and combine multiple prognostic factors while accounting for their interactions represents an ongoing area of investigation.

### 4.3. Non-Invasive Monitoring and Circulating Biomarkers

The development of circulating biomarkers that reflect disease biology and also predict treatment outcomes represents an important advancement in MDS monitoring [71,72]. The analysis of circulating microbial content has revealed distinct signatures associated with different disease subtypes and patient outcomes, suggesting their potential utility as prognostic biomarkers [71].

Previous studies examining microbial signatures in blood samples have identified evidence of dysbiosis in patients with MDS, with distinct patterns correlating with genetic mutations and clinical outcomes [71]. Although the mechanistic basis of these associations remains unclear, the potential of microbiome-based biomarkers adds another dimension to comprehensive patient assessments.

Liquid biopsy approaches using circulating tumor DNA and cell-free nucleic acids have been developed to monitor MDS. These non-invasive methods enable the real-time assessment of treatment responses, early detection of relapse, and monitoring of clonal evolution without repeated bone marrow procedures, although technical challenges related to the relatively low mutant allele frequencies in MDS remain to be overcome.

## 5. Clinical Implementation and Future Perspectives

### 5.1. Translating Complexity into Clinical Practice

The translation of sophisticated molecular insights into practical clinical improvements faces significant challenges related to test standardization, interpretation of results, and the development of treatment algorithms. The implementation of stem cell architecture-based treatment selection requires the standardization of flow cytometric approaches and the development of automated analysis platforms to ensure reproducible results across institutions.

Quality assurance programs and proficiency testing are essential to ensure that complex biomarker assessments are reliably performed in diverse clinical settings. The development of standardized protocols and interpretation guidelines represents a critical step toward the widespread clinical adoption of precision medicine approaches for MDS. Recent advances in flow cytometry standardization for minimal residual disease detection in acute leukemia have provided a framework for similar efforts in MDS [73].

The complexity of integrating multiple biomarkers necessitates the development of clinical decision-support systems that can process comprehensive molecular profiles and provide evidence-based treatment recommendations. However, such systems must be carefully validated and continuously updated as new evidence emerges, particularly given the rapid pace of discovery in the field of MDS biology.

### 5.2. Addressing Current Limitations and Future Research Priorities

Despite significant advances being made, several important limitations remain in the understanding and treatment of MDS. The biological basis of treatment resistance in specific patient subsets, particularly the mechanisms by which primitive stem cells survive therapeutic interventions, remains incompletely understood.

The cost and complexity of comprehensive molecular profiling pose practical barriers to its implementation, particularly in resource-limited settings. Developing streamlined approaches that can capture the most clinically relevant information while remaining practical for routine use is an important challenge.

Real-time monitoring approaches using liquid biopsy and other noninvasive techniques can enable dynamic treatment adjustments based on response patterns and emerging resistance mechanisms. These approaches could facilitate personalized and adaptive treatment strategies that can be modified based on evolving disease characteristics.

The development of composite endpoints that capture both clinical benefits and improvements in quality of life is important for demonstrating the value of new therapies [72,74,75]. Patient-reported outcome measures are becoming increasingly important components of clinical trial designs and treatment evaluations in patients with MDS.

Several key research priorities have emerged from the current advances in understanding MDS. The mechanistic connections between different molecular pathways, particularly the relationships between splicing factor mutations, stem cell architecture, and therapeutic vulnerabilities, require further elucidation.

Therefore, the development of combination therapies that can simultaneously target multiple resistance mechanisms is critical. A rational combination design based on biological understanding rather than empirical testing is essential for maximizing therapeutic benefits while minimizing toxicity.

Understanding the mechanisms that drive the transformation from low-risk to high-risk MDS, and eventually acute leukemia, is an important research priority [51,76]. The identification of early intervention strategies to prevent or delay disease progression can significantly improve patient outcomes.

However, the role of the microenvironment in disease progression and treatment resistance requires further investigation. The development of therapeutic approaches that can modify the bone marrow microenvironment to eliminate supportive niches for malignant stem cells is an underexplored opportunity.

## 6. Outstanding Research Questions and Future Directions

Despite remarkable progress in understanding MDS pathogenesis, several critical questions remain unanswered and represent priority areas for future investigation.

### 6.1. Mechanistic Integration Questions

The mechanisms linking splicing factor mutations to distinct stem cell architectures remain poorly understood. While we observe clear lineage biases in splicing factor effects, how these cell-type-specific consequences contribute to the formation of distinct CMP- or GMP-patterned architectures requires further elucidation.

### 6.2. Therapeutic Development Priorities

Critical questions for therapeutic development include the following: Can we identify key nodal points in the interconnected pathogenic network where single interventions can disrupt multiple mechanisms? How can we develop combination strategies that address primitive stem cell persistence, microenvironmental protection, and genetic complexity simultaneously?

The optimization of biomarker integration remains challenging. How can we develop clinical decision-making algorithms that appropriately weight multiple, sometimes conflicting, biomarker profiles? What are the minimum biomarker panels needed to guide effective treatment selection while remaining practical for routine clinical use?

### 6.3. Clinical Implementation Challenges

Practical questions for clinical implementation include the following: How can we standardize complex molecular profiling approaches across diverse clinical settings? What quality assurance programs are needed to ensure reliable biomarker assessments? How can we develop clinical decision-support systems that can process comprehensive molecular profiles and provide evidence-based treatment recommendations? The development of resistance monitoring strategies represents another critical area. Can liquid biopsy approaches enable the real-time assessment of treatment responses and early detection of resistance mechanisms? How can we implement adaptive treatment strategies that modify therapy based on evolving disease characteristics?

Addressing these questions will require coordinated efforts combining basic research, translational studies, and clinical trials designed to test systems-level hypotheses rather than single-mechanism approaches.

## 7. Conclusions

The field of MDS research has undergone a remarkable transformation over the past three years, characterized by the emergence of an integrated understanding of disease pathogenesis. The convergence of single-cell technologies, advanced genomics, and precision medicine approaches has revealed that MDS is best understood as a complex network of interconnected molecular mechanisms, rather than a collection of independent genetic and epigenetic alterations.

The identification of distinct stem cell architectures has provided a unifying framework for understanding disease heterogeneity and predicting therapeutic responses. (Nine stem cell architectures drive myelodysplastic syndrome progression and predict the response to venetoclax-based therapy.) And single-cell technologies have revealed the profound context-dependence of genetic alterations [7,8]. The discovery of epitranscriptomic regulation has added a new dimension to our understanding of post-transcriptional control [10], and the recognition of complex microenvironmental alterations has highlighted the importance of considering MDS as a disease affecting the entire bone marrow ecosystem [11,29].

These mechanistic insights are rapidly being translated into precise therapeutic approaches, with stem cell architecture-based venetoclax selection representing one of the first successful examples of biomarker-guided therapy for MDS [12,13]. However, the clinical implementation of precision medicine in MDS faces significant challenges related to the complexity of integrating multiple biomarkers, the cost and logistics of comprehensive molecular profiling, and also the need for standardized approaches that can be reliably implemented across diverse clinical settings.

The most significant challenge is the development of clinical decision-making algorithms that can appropriately integrate multiple, sometimes conflicting, biomarker profiles into actionable treatment recommendations. Patients with favorable stem cell architectures but adverse genetic features represent a common clinical scenario that requires careful consideration of competing prognostic factors. Recent clinical data have highlighted the dominant impact of certain adverse genetic alterations such as TP53 mutations, which may override other potentially favorable biomarkers [25,37].

An emerging systems-level understanding of MDS pathogenesis is driving the development of more sophisticated therapeutic approaches that can simultaneously target multiple interconnected pathways. The failure of single-agent approaches in many patients suggests that combination strategies designed to address the networked nature of MDS biology are required to achieve a deep and durable response.

This review presents several key contributions to the understanding of MDS pathogenesis and treatment. First, we have demonstrated that MDS is best understood as a complex network of interconnected molecular mechanisms rather than a collection of independent genetic and epigenetic alterations. The integration of splicing factor mutations, stem cell architecture alterations, epitranscriptomic dysregulation, and microenvironmental changes creates a unified pathogenic framework that explains disease heterogeneity and predicts therapeutic responses. Second, we have shown how mechanistic insights can be successfully translated into precision therapeutic approaches, with stem cell architecture-based venetoclax selection representing a paradigm for biomarker-guided therapy in MDS. Third, we have identified the critical challenges in implementing precision medicine approaches and proposed practical solutions for translating complex molecular insights into clinical improvements.

The convergence of technological innovations, mechanistic understanding, and therapeutic development has positioned MDS research at the forefront of precision oncology. The systems-level perspective presented in this review offers a roadmap for developing more effective therapeutic strategies that address the networked nature of MDS biology. While significant challenges remain in standardizing complex molecular profiling and developing integrated clinical decision-making algorithms, the framework presented here provides a foundation for the next generation of MDS research and clinical care. Future progress will depend on our ability to maintain this integrated perspective while developing practical tools that can translate biological complexity into improved patient outcomes.

## Figures and Tables

**Figure 1 genes-16-00928-f001:**
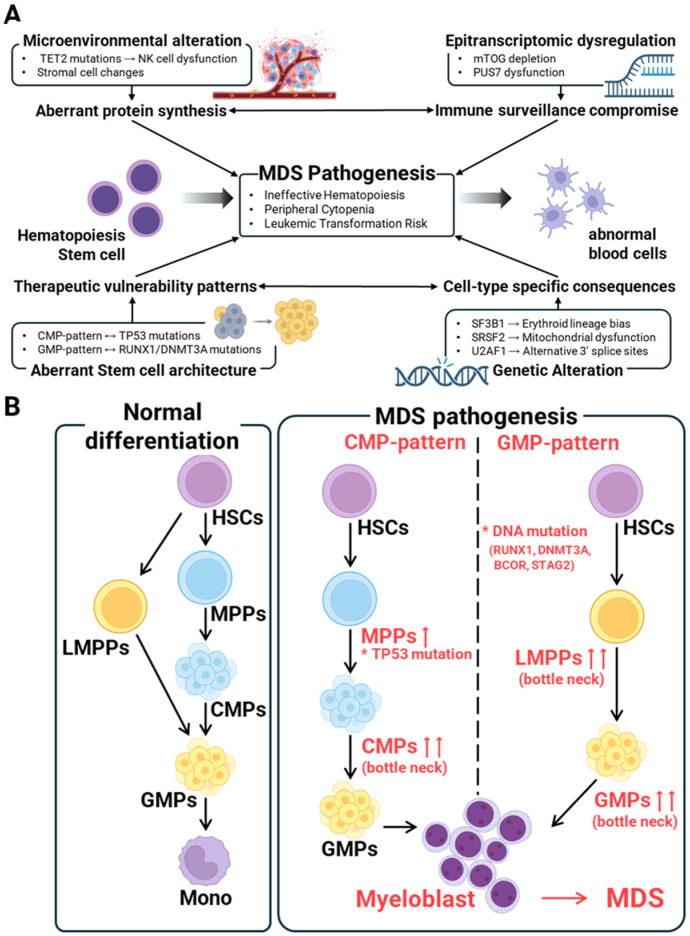
An integrated model of MDS pathogenesis and its link to aberrant stem cell architecture. (**A**) An integrated view of MDS pathogenesis. This panel illustrates the modern, integrated understanding of MDS. It shows how four key pathogenic pillars—genetic alterations (e.g., splicing factor mutations), aberrant stem cell architecture, epitranscriptomic dysregulation, and microenvironmental alterations—form an interconnected network. These pathways collectively drive the clinical phenotype of ineffective hematopoiesis and the risk of leukemic transformation. (**B**) Stem cell architecture defines MDS heterogeneity. This panel provides a detailed view of a key mechanism from panel A. It contrasts normal hematopoiesis with the two main MDS stem cell architectures that contribute to the disease’s heterogeneity. The CMP pattern shows a distinct expansion of common myeloid progenitors and is frequently associated with *TP53* mutations. In contrast, the GMP pattern is characterized by an expansion of granulocyte–monocyte progenitors and is linked to mutations such as *RUNX1* and *DNMT3A*.

**Figure 2 genes-16-00928-f002:**
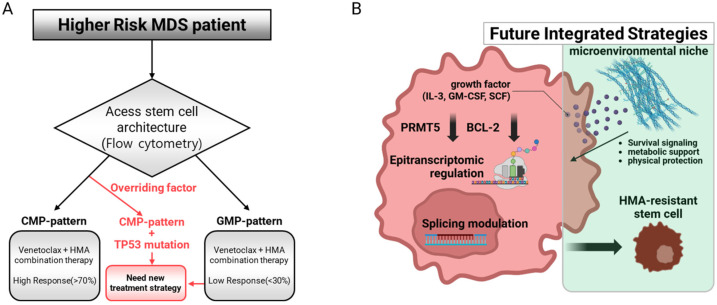
Precision therapeutic strategies and the evolving therapeutic landscape in MDS. (**A**) Biomarker-guided therapy for venetoclax. This flowchart depicts a precision medicine strategy in MDS where biological understanding directly informs treatment selection. Patient stem cell architecture is used to predict the response to venetoclax-based therapy. Patients with a CMP pattern achieve markedly superior response rates (>70%), while patients with GMP-pattern MDS respond poorly (<30%). However, the presence of a TP53 mutation is a dominant negative prognostic factor that significantly diminishes treatment efficacy, overriding the predictive value of the stem cell architecture. (**B**) Novel therapeutic targets and resistance mechanisms. This diagram illustrates emerging therapeutic opportunities and challenges. Novel intracellular targets include splicing modulators for up to 60% of MDS patients with splicing factor mutations, and PRMT5 inhibitors exploit synthetic lethality to target these mutations. Key resistance mechanisms are also shown, including the failure of HMA therapy to eliminate primitive, disease-initiating stem cells and the protective niche provided by the bone marrow microenvironment, which shields malignant cells from therapeutic agents.

**Table 1 genes-16-00928-t001:** Novel mechanisms in MDS understanding and precision therapeutic approaches. This table summarizes four key molecular mechanisms that have emerged as critical drivers of MDS pathogenesis, each demonstrating distinct cell-type specificities and clinical manifestations that inform precision therapeutic strategies. These mechanisms represent recent advances in understanding MDS biology and provide actionable targets for personalized treatment approaches.

Molecular Mechanism	Cell-Type Specificity	Key Clinical Manifestations	Therapeutic Implications	Ref
**Splicing Factor Mutations** (SF3B1, SRSF2, U2AF1)	SF3B1: Ring sideroblast formation megakaryocytic-erythroid preference [7,17] SRSF2: Mitochondrial dysfunction, multiple lineages [20,22] U2AF1: Distinct 3′ splice site patterns [23]	Ring sideroblasts (>15%) [7] macrocytic anemia [17] lineage-specific dysplasia [8] iron overload [19]	PRMT5 inhibition synthetic lethality [21] H3B-8800 splicing modulators in trials [22] personalized mutation-based therapy [23]	[7,8,17,19,21,22,23]
**Stem Cell** **Architecture Alterations**	CMP-pattern: Increased CD34+CD38+CD123+ progenitors [9] GMP-pattern: Expanded CD34+CD38+CD123- progenitors [9]	CMP-pattern: TP53 mutations [14,25] GMP-pattern: RUNX1/DNMT3A Mutations [15,16]	Venetoclax response prediction >70% (CMP) vs. <30% (GMP) [9] guides combination therapy selection [12,33]	[9,12,14,15,16,25,33]
**Epitranscriptomic Dysregulation** (Pseudouridine modifications)	CD34+ HSPCs through mTOG-mediated translational control [10,27] ribosome dysfunction [27]	Aberrant protein synthesis [10] impaired differentiation [27] increased transformation risk [10]	PUS7 inhibition [27] mTOG pathway targeting [10] proteostasis modulators [27]	[10,27]
**Microenvironmental Alterations**	TET2 mutations affect NK cells [29] stromal reprogramming [11,28] cytokine dysregulation [32]	Immune evasion [29] impaired normal hematopoiesis [11] chronic inflammation [32]	Immune checkpoint inhibitors + HMA [29] NK cell therapy [29] stromal targeting [11,32]	[11,28,29,32]

**Table 2 genes-16-00928-t002:** Comprehensive analysis of treatment resistance mechanisms in MDS and therapeutic countermeasures. This table systematically categorizes the major mechanisms underlying treatment failure in MDS, detailing their molecular basis, clinical impact, current management approaches, and emerging therapeutic strategies. The interconnected nature of these resistance mechanisms underscores the need for combination approaches that target multiple pathways simultaneously, as discussed in the accompanying text.

Resistance Mechanism	Primitive Stem Cell Persistence	Microenvironment-Mediated Protection	Dominant Genetic Alterations
**Molecular/Cellular Basis**	Maintained self-renewal in disease-initiating cells [9,11]	Stromal cell survival signals [11,28] drug metabolism barriers [32]	TP53 biallelic mutations [14,25]
**Clinical Manifestation**	Initial response followed by relapse [12,33]	Variable response rates [35]	Refractory disease regardless of biomarkers [14,50]
**Current Therapeutic Approaches**	Intensified HMA dosing extended treatment [3,5]	Standard combination regimens [12,13]	Early transplant consideration [14]
**Emerging Strategies**	Stem cell architecture-guided therapy [9]	Niche-disrupting agents [11,32]	p53 pathway restoration [50]
**Ref**	[3,5,9,12,33]	[11,12,13,28,32,35]	[14,25,50]
**Resistance Mechanism**	**Adaptive Clonal Evolution**	**Multi-pathway Dysfunction**	**Epitranscriptomic Resistance**
**Molecular/Cellular Basis**	Selection pressure-driven mutation acquisition [48,51,52]	Complex genetic landscapes [53,54]	Dysregulated protein synthesis machinery [10,27]
**Clinical Manifestation**	Progressive treatment failure [48]	Poor single-agent response [35,48]	Intrinsic treatment resistance [10]
**Current Therapeutic Approaches**	Sequential therapy changes [35]	Empirical combination approaches [12,37]	No specific targeting available
**Emerging Strategies**	Dynamic treatment algorithms [48]	Personalized multi-target strategies [36]	PUS7-mTOG pathway modulators [10,27]
**Ref**	[35,48,51,52]	[12,35,36,37,48,53,54]	[10,27]

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
