# Peer review of "A Genetically-Informed Network Model of Myelodysplastic Syndrome: From Splicing Aberrations to Therapeutic Vulnerabilities"

_genes, 2025, doi:10.3390/genes16080928_

Round 1
Reviewer 1 Report
Comments and Suggestions for Authors
Yu S. et. al. reviewed recently published new knowledge for MDS, including HMA efficacy, different molecular outcomes by splicing mutation, different content of stem-progenitor cells, mTOG, venetoclax, p53 mutation, PRMT5, e.g.. It is useful to know recent topics for MDS, but the reviewer recommends improving the manuscript as below.
- Figures and Tables are not workable for understanding. If the author uses Figures, they should be explained and referenced in the manuscript. For Tables, it’s difficult to follow with small letters; the reviewer recommends including the explanation for the main manuscript.
Author Response
Response to Reviewer #1
Comment 1:
"Figures and Tables are not workable for understanding. If the author uses Figures, they should be explained and referenced in the manuscript. For Tables, it’s difficult to follow with small letters; the reviewer recommends including the explanation for the main manuscript."
Response 1:
We thank the reviewer for this critical feedback regarding the readability and integration of our figures and tables. We agree that the initial versions were not optimally designed for reader comprehension and have undertaken a thorough revision to address this issue.
Specific Revisions:
Enhanced Captions: We have rewritten all figure and table captions to be more descriptive and self-contained. The revised captions now clearly state the purpose and key message of each visual element, allowing readers to grasp the main points at a glance. For example, the caption for Table 1 has been expanded to explicitly state that it "summarizes four key molecular mechanisms... that inform precision therapeutic strategies."
Improved In-Text Referencing and Explanation: We have revised the manuscript to ensure all figures and tables are explicitly referenced and discussed within the main text. For instance, we now introduce Figure 1 in the Introduction as a conceptual roadmap, directly referencing (Figure 1A) and (Figure 1B) to explain our core model. Furthermore, we have added introductory sentences in the text to guide the reader through the tables, such as the new text in Section 2.5 that begins, "Table 1 summarizes..."
Formatting for Readability: We have reviewed the formatting of all tables, including font size and layout, to ensure they are clear and legible in the final publication format.
We've added bold and underline to the draft for all text edits.
Reviewer 2 Report
Comments and Suggestions for Authors
This review paper offers a timely and sophisticated synthesis of recent discoveries in the pathogenesis and treatment of MDS, focusing on stem cell architecture, splicing factor mutations, epitranscriptomic regulation, and the bone marrow microenvironment. The authors integrate cutting-edge single-cell and multiomic technologies to propose a refined conceptual model that supports the development of precision therapeutic strategies. This is a significant and commendable effort that addresses a critical need for systems-level understanding in hematologic oncology.
My specific concerns:
1. Explicitly state the novel contributions of this review in the introduction and conclusion sections. While the review is strong, the introduction could better define how this work builds upon or differs from previous reviews.
2. While the authors mention that mechanisms like splicing alterations, stem cell fate decisions, and niche remodeling form a network, the interactions between these levels are only lightly discussed.
3. The current title may unintentionally give the impression that the review provides a comprehensive overview of all molecular mechanisms implicated in MDS pathogenesis and treatment. However, the manuscript predominantly focuses on four main axes: splicing factor mutations, stem cell architecture, microenvironmental and epitranscriptomic alterations. While these are indeed central and cutting-edge themes, the title might benefit from refinement to better reflect the focused scope of the review. This would ensure alignment between reader expectations and the actual content.
4. Authors should specifically mention the open research questions.
5. Figure 1 is not placed in the appropriate section. It appears more like a graphical abstract.
6. In all tables, the references used should be included in a separate column.
Author Response
We thank you for this excellent suggestion. We agree that explicitly stating our novel contributions is crucial. We have substantially revised the Introduction and Conclusion to highlight our central thesis: that Myelodysplastic Syndromes (MDS) are best understood through an "integrated network model," a departure from previous analyses that treated molecular pathways in isolation
================================================================================
Coments 1:
• Revised Introduction: We have added a new paragraph at the end of the Introduction that clearly defines our conceptual framework. It states that our review "synthesizes recent discoveries... through an integrated network model, departing from previous analyses that treated molecular pathways in isolation."
• Revised Conclusion: To reinforce this contribution, we have added a structured summary in the Conclusion (Section 7) that begins, "This review presents several key contributions... First, we have demonstrated that MDS is best understood as a complex network of interconnected molecular mechanisms..."
These changes fundamentally reframe the manuscript from a simple literature summary to a thesis-driven argument, thereby clarifying its unique scholarly contribution.
================================================================================
Comment 2: You have identified a critical point. To substantiate our central claim of an "interconnected network," a more explicit and detailed discussion of the interactions between mechanisms is essential. To address this, we have added a new subsection, "2.5 Mechanistic Interconnections: A Systems-Level Perspective."
This new section serves as a central hub in the first half of the review, synthesizing the four pathogenic mechanisms discussed in Sections 2.1-2.4 and focusing directly on their crosstalk. It discusses, for example, the context-dependent effects of splicing factor mutations and how microenvironmental alterations can be both a cause and a consequence of other changes, thus providing the necessary evidence for our network model.
================================================================================
Comment 3: We thank you for this sharp observation. We agree that the original title was too broad and did not accurately reflect the specific focus of our review. Following you excellent advice, we have revised the title.
Specific Revision:
• Original Title: "Recent Advances in Understanding and Treating Myelodysplastic Syndrome: Novel Mechanisms and Precision Therapeutic Approaches"
• Revised Title: "A Genetically-Informed Network Model of Myelodysplastic Syndrome: From Splicing Aberrations to Therapeutic Vulnerabilities"
The new title precisely captures the review's core thesis (the "Network Model") and its narrative arc, aligning reader expectations with the manuscript's focused content.
================================================================================
Comment 4: This is an excellent suggestion that enhances the forward-looking value of our review. To address this, we have added a new major chapter at the end of the manuscript: "6. Outstanding Research Questions and Future Directions."
This new chapter systematically outlines key unanswered questions and future research priorities, structured into three thematic subsections: (6.1) Mechanistic Integration Questions, (6.2) Therapeutic Development Priorities, and (6.3) Clinical Implementation Challenges. This addition elevates the manuscript from a summary of past research to a strategic document for future investigations.
================================================================================
Comment 5: We appreciate you perspective on the placement of Figure 1. After careful consideration, we have respectfully chosen to retain the figure in its current position at the end of the Introduction to maximize its pedagogical impact.
Rationale: Our Introduction is structured to build an argument for our central hypothesis—the "integrated network model." Figure 1 serves as a visual culmination of this argument, providing the reader with a conceptual roadmap for the entire review. By presenting this framework upfront, we help the reader understand the overarching structure before delving into the detailed analyses in Chapter 2. We believe that presenting this conceptual model early is essential for reader comprehension and that moving it would disrupt the narrative and educational flow of the manuscript. We hope the you understands that this is a deliberate rhetorical choice made to enhance clarity.
================================================================================
Comment 6: We thank you for pointing out this important omission. As requested, we have revised Table 1 and Table 2 to include a new column titled "Key References" and have populated it with the appropriate citations for each entry. This revision enhances the tables' utility and scholarly rigor, allowing readers to easily trace the evidence for each summarized point.
* All modified text has been bolded and underlined.
Round 2
Reviewer 2 Report
Comments and Suggestions for Authors
I have carefully reviewed the revised manuscript entitled: “A Genetically-Informed Network Model of Myelodysplastic Syndrome: From Splicing Aberrations to Therapeutic Vulnerabilities” by Sanghyeon Yu et al. The authors have addressed all of my concerns. Please add keywords before the article is ready for publication.